# Small-Scale Mechanical Testing of Cemented Carbides from the Micro- to the Nano-Level: A Review

**Annamária Naughton-Duszová** [1] , **Tamás Csanádi** [2], **Richard Sedlák** [2], **Pavol Hvizdoš** [2,]* and **Ján Dusza** [2]

1   Lukasiewicz Research Network—The Institute of Advanced Manufacturing Technology, Wrocławska 37a, 30–011 Krakow, Poland; annamaria.duszova@ios.krakow.pl
2   Institute of Materials Research, Slovak Academy of Sciences, Watsonova 47, 04001 Košice, Slovak Republic; tcsanadi@saske.sk (T.C.); rsedlak@saske.sk (R.S.); jdusza@saske.sk (J.D.)
*   Correspondence: phvizdos@saske.sk; Tel.: +421-55-792-2402

**Abstract:** In this overview, we summarize the results published to date concerning the small-scale mechanical testing of WC–Co cemented carbides and similar hardmetals, describing the clear trend in the research towards ever-smaller scales (currently at the nano-level). The load-size effect during micro/nanohardness testing of hardmetals and their constituents and the influence of the WC grain orientation on their deformation, hardness, indentation modulus, fracture toughness, and fatigue characteristics are discussed. The effect of the WC grain size/orientation, cobalt content, and testing environment on damage accumulation, wear mechanisms, and wear parameters are summarized. The deformation and fracture characteristics and mechanical properties, such as the yield and compression strength, of WC–Co composites and their individual WC grains at different orientations during micropillar compression tests are described. The mechanical and fracture properties of micro-cantilevers milled from WC–Co hardmetals, single WC grains, and cantilevers containing WC/WC boundaries with differently-oriented WC grains are discussed. The physical background of the deformation and damage mechanisms in cemented carbides at the micro/nano-levels is descri and potential directions for future research in this field are outlined.

**Keywords:** cemented carbides; WC-Co; micro/nano mechanics; damage; characterization

## 1. Introduction

Cemented carbides, or hardmetals, are widely used for extremely demanding applications, such as cutting, forming, and mining tools and even as structural elements due to their hardness and strength, fracture toughness, and excellent wear resistance. This is a direct consequence of their complex composite structure of interpenetrating networks of a hard carbide phase, usually tungsten carbide, and a tough metallic binder, usually cobalt with dissolved carbon and tungsten. The outstanding intrinsic properties of these phases, the less brittle character of WC in comparison with other monocarbides, the fcc–hcp phase transformation in the binder (providing an additional micromechanism for damage tolerance), and the synergic effects of the two phases' combined response, which results in optimal interface characteristics, are behind the success of these composites [1–3].

These materials' mechanical properties are determined by processing flaws and microstructure parameters. The bending strength increases with decreasing flaw size, the fracture toughness with increasing mean free path in the binder phase, and the hardness with increasing contiguity (increasing WC/WC interface) [1–5]. The influence of different processing flaws, which serve as fracture origins, on the bending strength and their scatter (Weibull modulus) and the influence of microstructure

parameters on the hardness, fracture toughness, fatigue, tribology, compressive strength, and creep have been investigated intensively in recent decades [4–10].

The evaluation and understanding of strength properties of cemented carbides in relation to inherent flaws arising from processing is generally implemented on the basis of linear-elastic fracture mechanics (LEFM) and fractography, under the principle that mechanical failure involves propagation of pre-existing defects [4,6].

Different methods have been applied for measuring fracture toughness during the last 30 years and several micromechanical models have been developed and used to study crack propagation phenomena in hardmetals based on the plastic deformation of the binder phase, the contiguity, shape, and size distribution of the WC grains and even the anisotropy of WC [10–14].

The microstructural aspects of the hardness, fatigue, creep, scratch/wear characteristics, deformation, and damage accumulation in WC–Co cemented carbides at room and high temperature and how they relate to their microstructure parameters have been investigated using different indentation, compression and bending tests, scratch/tribological testing, etc. [7,8,15,16].

It has been found that the main role of WC (or other carbides) in the microstructure of hardmetals is to provide hardness and wear resistance. On the other hand, the constrained plastic deformation and rupture of the tough binder ligaments, as well as the fracture of the carbide-binder interfaces, mainly accounts for the global fracture toughness of cemented carbides. Although rupture of single WC grains is believed to be only a minor contribution to macroscopic failure, inter-crystalline fracture is considered to be an important factor and evaluation of the effects of crystal anisotropy on toughness in WC crystals may also be important for understanding the global mechanical response of WC–Co hardmetals.

Most of these investigations have been carried out at the macro scale on specimens with relatively large volumes, applying tests under compression and bending loads (static, dynamic, fatigue modes) at room and high temperatures in combination with a wide range of microstructural characterization and analytical methods.

Recently, novel hardmetals with tailored composition, microstructure, and optimized properties have been developed and are under development for various advanced applications. These new systems are: (i) Near-nano- and nano-structured hardmetals; (ii) ultra-coarse hardmetals with nano-grain-reinforced binder; (iii) functionally-graded hardmetals; (iv) hardmetals with alternative binder phases, such as Fe-Ni/Fe-Ni-Co alloys or high-entropy alloy (HEA) phases; (v) and textured hardmetals with highly-oriented WC grains, among others [17–20].

During the development of such new types of hardmetals, new testing and characterization methods were applied, including micro- and nanomechanical testing methods, in addition to the traditional macromechanical techniques.

Micro- and nanomechanical testing have undergone rapid development over the last decade, with miniaturized test rigs and MEMS-based devices providing access to the mechanical, deformation, and fracture properties of materials, including hardmetals, on length scales from the micrometer down to tenths of a nanometer [21]. Testing of materials at small scales is very important because the mechanical failure of any bulk material starts with the formation, extension, or local accumulation of initially small defects, leading finally to a catastrophic fracture by an expanding crack. Thus, any bulk material profits from an in-depth understanding of its deformation and mechanical phenomena at the nano- and micrometer length scale.

The first, and still the most often used, of the micro/nanomechanical testing methods is the indentation test, widely used over the last few decades as a fast, simple, and nondestructive method for measuring the mechanical properties of a variety of materials [22–25]. Indentation tests, in the form of classical or instrumental hardness tests, applied at different indentation loads, can cover the whole range of material testing scales, from macroscale to nanoscale—see Figure 1 for a depiction of these tests as applied to hardmetals.

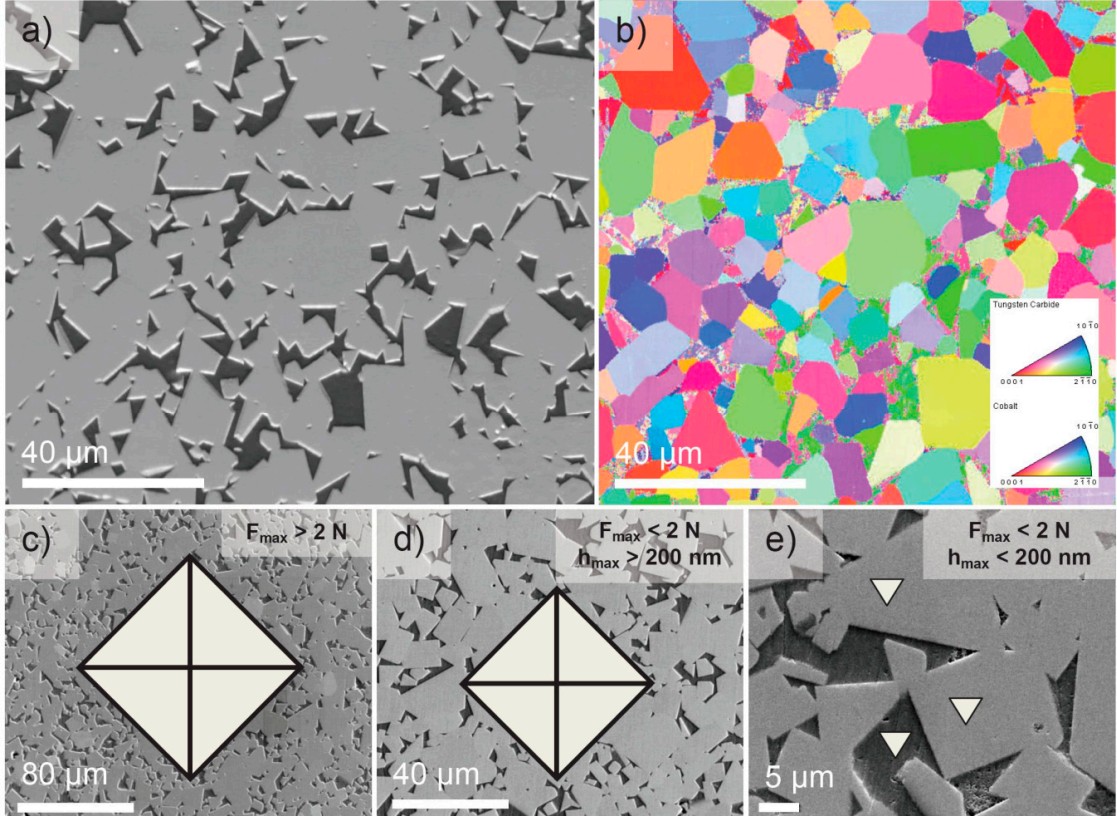

**Figure 1.** Characteristic microstructure of the system WC–Co, as imaged by (**a**) scanning electron microscopy (SEM) and (**b**) electron back scattering diffraction (EBSD). Schematic illustration of the hardness testing at different levels with $F_{max}$ as the maximum test force and $h_{max}$ as the maximum indentation depth. (**c**) Macro-range: $F_{max} > 2$ N, (**d**) micro-range: $h_{max} > 200$ nm and $F_{max} < 2$ N, and (**e**) nano-range: $h_{max} \leq 200$ nm, after [25].

Micro- and nanomechanical testing methods started to evolve with the invention of nanoindentation and scanning probe microscopy (SPM) techniques and since then nanoindentation has frequently been used to investigate both bulk materials and coatings [26]. Over the last few decades, nanoindentation instruments have improved their capacity to obtain highly localized and accurate information on mechanical properties. Additionally, they have expanded the range of test techniques beyond simple nanoindentation, with several including some capability for nanotribological measurements (e.g., nano-scratch and nano-wear testing, high strain rate tests, etc.) which has consequently greatly expanded their range of applications [27,28].

The advent of micro-pillar compression began with the work of Uchic et al. [29], who sought to probe the mechanical performance of materials at small length scales under uniaxial stress conditions, thus avoiding the complications of performing nanoindentation testing with a multiaxial stress state below the indenter tip. In their study, focused ion beam (FIB) milling was applied to prepare micron-sized pillars, of varying dimensions, of the materials under study; these were subsequently deformed by using a nanoindenter equipped with a flat punch, widening out the experimental capabilities of micro/nanomechanical testing. This methodology possesses many advantages, but also some shortcomings, connected with (i) possible alteration of the mechanical properties by the FIB milling procedure, (ii) the controlled stress state during the test, and (iii) imperfections of the instrument [30–32].

Some of these shortcomings can be avoided by micro-tensile testing, as introduced by Kiener [33]. In micro-tensile testing, the mechanical behavior of a material is determined under uniaxial loading

using a miniaturized specimen and the strain generated in the specimen during loading remains uniform over its thickness; the results obtained are easy to interpret [34–36].

The micro-beam bending method involves bending free-standing micro-cantilevers to fracture using wedge or atomic force microscopy (AFM) tips attached to a nanoindenter. The cantilever beams can be of different configurations—single, double, clamped, and notched—and their deformation can be approximated using formulae from simple beam bending theory. The analytical solutions are very sensitive to the beam dimensions, which must be accurately measured [37–39].

In recent years, the advantages of nanomechanics have been increased further still by integrating the miniaturized testing systems mentioned above with various characterization techniques, permitting measurement of the evolution of local strain, sub-structural defects, and the interactions and kinetics of defects. SEM, transmission electron microscopy (TEM), and scanning TEM (STEM) constitute extremely powerful complementary techniques for in situ micro- and nanomechanics, with their high spatial resolution and capacity to produce images and record diffraction patterns. Recently, improved resolution of defects (e.g., dislocations, stacking faults) by SEM has been provided by electron channeling contrast imaging (ECCI), supported by electron back scattering diffraction (EBSD) [40–44].

In addition to room-temperature nanomechanical testing, testing at high temperatures in both ex situ and in situ modes has undergone intensive development and been increasingly widely used over the last decade. In order to perform quantitative nano/micromechanical testing at elevated temperatures, numerous matters concerning the sample, testing rig, and indenter tip/flat punch material (among others) must be carefully chosen or monitored in order to prevent chemical alterations of the tested material and ensure fine control of thermal gradients of the parts, among other issues which must be addressed [45–48].

Methods of testing at such a small scale present many challenges with respect to positioning and handling of the specimen, as well as accurate application of force and reliable measurement of load and displacement during deformation. The choice of method generally depends on the property to be measured, but also on the difficulties of fabrication, instrumentation, and testing and on the manner in which data interpretation, analysis, and parameter extraction must be performed.

In micro/nanomechanical testing the preparation of the specimen is extremely important. A variety of advanced micro-fabrication methods, such as electron discharge machining (EDM), focused ion beam (FIB) machining, computerized numerically controlled (CNC) machining, laser-based processes, lithography, galvanoforming and plastic molding, and electro-deposition have been adopted for fabrication of specimens at the micrometer and sub-micrometer length scales suitable for testing by the methods mentioned above [49–51].

Micro- and nanomechanical testing of brittle materials, such as advanced ceramics and hardmetals, have undergone rapid development over the last decade, providing access to the mechanical properties and performance of materials on length scales from a micrometer down to tenths of a nanometer [52–63]. The non-elastic mechanical behavior of ceramics, which are inherently brittle, is difficult to measure at room temperature due to their very limited ductility. However, the plastic characteristics of ceramics are very important and can be tested at the micro- to submicron level (the level of the grain size) where the probability of flaws is significantly reduced. Nanomechanical tests are also very important here, because accurate prediction of a material's response also requires an understanding of the fundamental mechanisms of material deformation and fracture on the micro- and nanoscale. In the case of brittle materials, where deformation at the macroscale is strongly limited, nanomechanical testing is a promising way to achieve this.

The small-scale testing methods deployed during the last decade in the field of hardmetals research are illustrated in Figure 2.

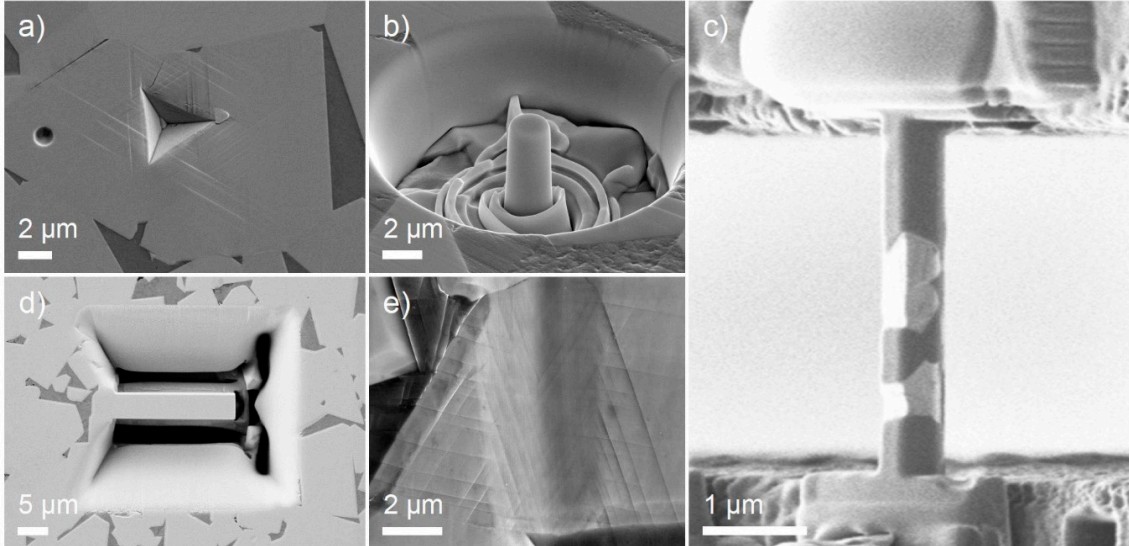

**Figure 2.** Most frequently used micro/nanomechanical testing methods in hardmetals research during the last decade: (**a**) Micro/nanoindentation, (**b**) micropillar compression, (**c**) micro-tensile testing (by courtesy of Namazu et al. [64] with copyright permission from Elsevier, 2015), (**d**) micro-cantilever bending, and (**e**) micro/nano-scratch testing.

In this overview of the field, we summarize the state of the art research concerning mechanical properties like hardness, strength, fracture toughness, tribological behavior, and deformation and damage and fracture characteristics of bulk cemented carbides and their constituents at the micro- and nanoscale.

## 2. Micro/Nanoindentation

Micro/nanoindentation testing has been frequently applied to characterize hardmetals in recent decades, with increasingly intense research in the field of nanoscale indentation testing.

The influence of the composition and microstructure parameters on microhardness and indentation modulus (elastic property of an anisotropic material derived from nanoindentation) has been frequently investigated, with very similar results to the case of macroscale hardness tests—the hardness increases with the volume fraction of WC and decreases with WC grain size and contiguity [65,66].

Deformation-induced microstructural changes can result in changes in the mechanical properties of these composites. During an indentation fatigue test of two WC–Co systems with different microstructure parameters, it was found that the system with the higher volume fraction of Co deforms more during the first cycle than the system with lower cobalt content (as expected). The changes in the microstructure of the systems in the damage zone below the indenter after the first cycle were studied. It was found that, in the system with lower volume fraction of Co, the WC/WC bridges were destroyed, resulting in decreased deformation resistance of the system. In the system with the higher cobalt content, a martensitic fcc–hcp phase transformation took place, resulting in formation of an hcp-Co phase with limited deformation ability and increased deformation resistance. As a result of these mechanisms, the system which deformed less during the first cycle exhibited significantly higher deformation during cyclic loading than the system which deformed more during the first cycle (Figure 3).

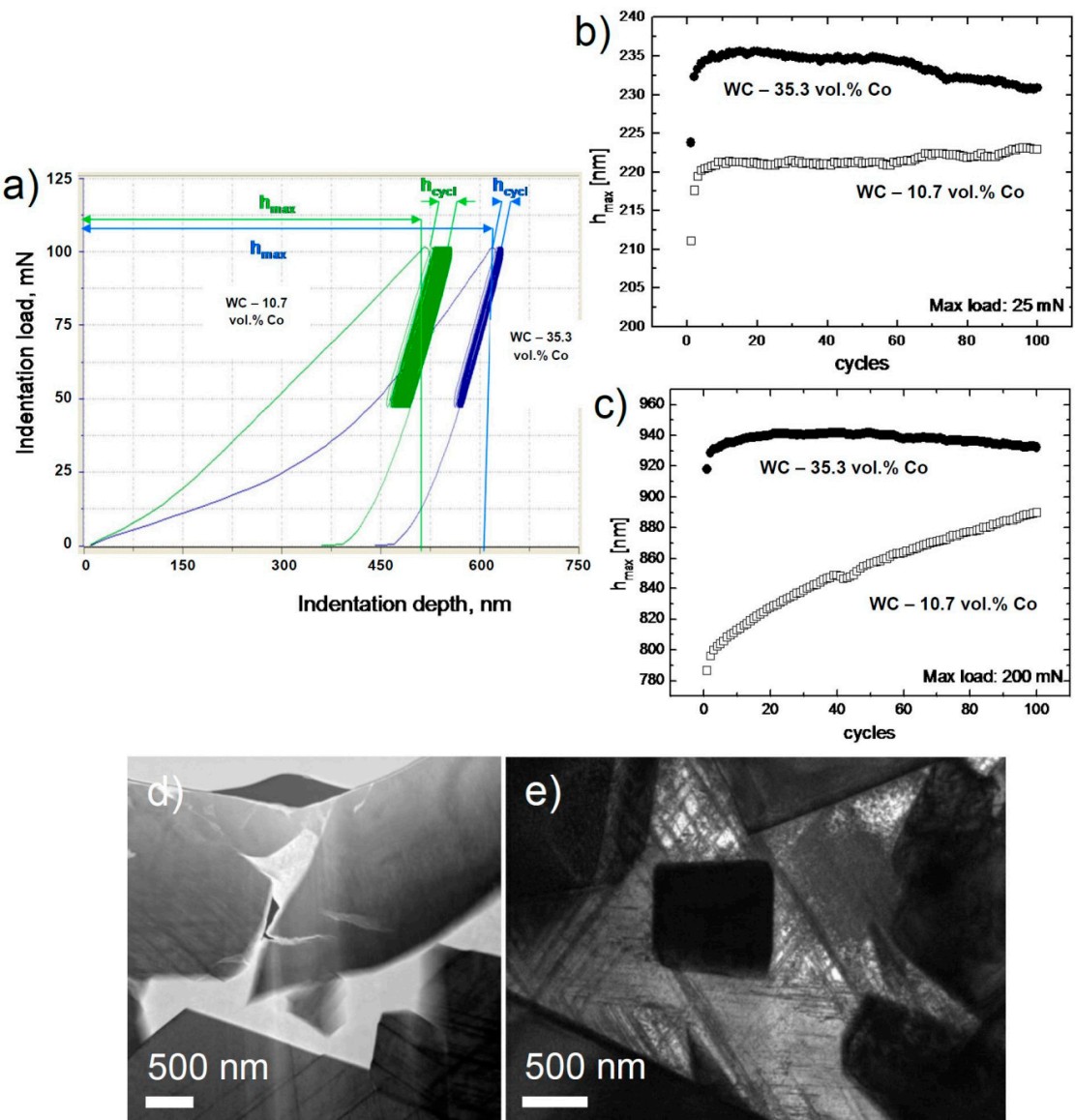

**Figure 3.** Characteristic indentation load vs. indentation depth relationship for two investigated systems with different cobalt contents. (**a**) Indentation depth values during the first cycle ($h_{max}$) and after 100 cycles; influence of the number of cycles on the depth of impression under a maximum indentation load of (**b**) 25 mN and (**c**) 200 mN; (**d**) TEM micrograph of the damaged zone just beneath the indenter tip after cyclic fatigue test (200 mN/100 cycles) in WC–Co with lower Co content; (**e**) TEM micrograph showing the structure of the deformed binder of the damaged zone just beneath the indenter tip after cyclic fatigue test (200 mN/100 cycles) in WC–Co with higher Co content. Reproduced from [67], with copyright permission from Elsevier, 2013.

It is well known that hardness varies (usually, increases) with decreasing load or indentation depth for many materials; this phenomenon has been called the indentation size effect (ISE). Several theories have been proposed to explain the ISE, of which probably the most popular is the gradient plasticity theory, wherein geometrically necessary dislocations (GNDs) are generated under the indenter due to strain gradients [68–70]. Decreasing the indentation size results in a greater density of GNDs and hence in higher measured hardness. According to this theory, ISE can be described as $H/H_0 = \sqrt{[1 + (h^*/h)]}$, where $H_0$ is the intrinsic hardness and $h^*$ is the characteristic length of the material.

The ISE was also analyzed in the case of cemented carbides, using different micro- and nanoindentation methods, and the results were similar to those for advanced ceramics.

The size effect in the case of cemented carbides with varying volume fraction of binder phase and WC grain size is shown in Figure 4a. It can be seen for all systems that the hardness decreases with increasing indentation depth and that this effect is greater for systems with greater hardness.

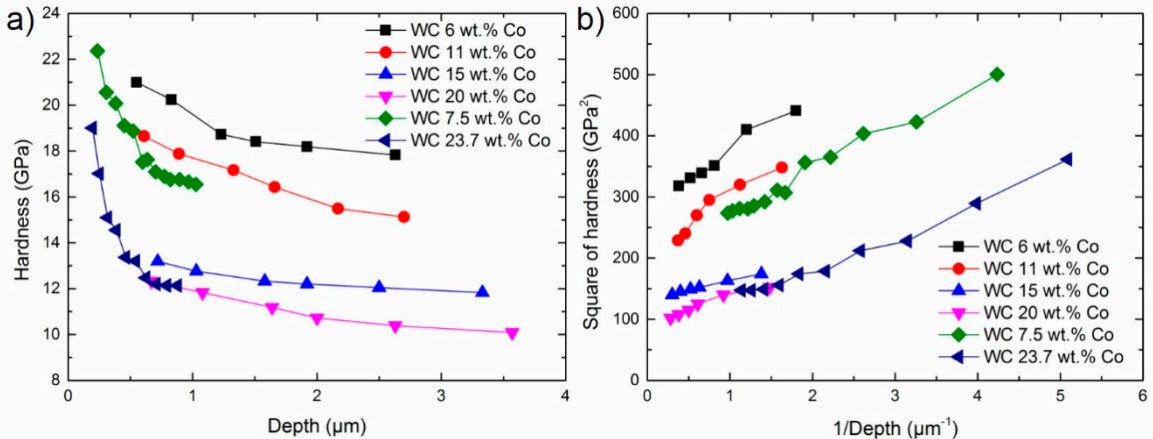

**Figure 4.** Load-size effect in WC–Co hardmetals with different microstructure parameters—which are defined by the different weight percent of Co: (**a**) Relationship between the indentation depth and hardness, (**b**) Square of hardness as a function of reciprocal depth. Reproduced from [65,69], with copyright permission from Taylor & Francis, 2006.

According to the aforementioned theory, if a material has an intrinsic length scale, it is to be expected that a plot of $H^2$ against $1/h$ will no longer be a straight line but will show an anomaly when $h$ is of the order of this intrinsic length. The WC–Co composite, consisting of grains of hard WC cemented together with relatively soft cobalt, is such a material and should show such behavior. The load-size effect of different grades of WC–Co systems, with different microstructure parameters, is shown in Figure 4. According to the results, these do not behave in agreement with this theory.

A nanoindentation test on WC grains in a WC–Co system found a clear load-size effect at the nano-level for loads up to 1 mN, with a more visible load-size effect at higher hardness values (Figure 5a) [71]. However, the indentation load size effect (ISE) is negligible for the binder phase, with an average hardness value of around 8 GPa.

The indentation loads involved were suitable for measuring the hardness of the WC grains; however, the measurement of the Co hardness without the influence of the surrounding WC is problematic, even at these loads (Figure 5b). In addition to the influence of the load, the crystallographic orientation of WC crystals was found to have a significant influence on their hardness and indentation modulus. To minimize surface effects, testing was also carried out with a higher load (10 mN) in order to measure and statistically evaluate the influence of the crystallographic orientation of the WC grains on hardness. The average hardness and indentation modulus of the basal planes and prismatic planes at the 10 mN load were: $HV_{basal} = 40$ GPa, $E_{basal} = 675$ GPa, $HV_{prismatic} = 33$ GPa, and $E_{prismatic} = 540$ GPa.

In addition to ISE, the hardness anisotropy (as shown in Figure 5) has also been widely studied, firstly using macro- and micro-indentation on WC single crystals and later by nanoindentation of WC grains in WC–Co. A list of these works is given in Table 1 [71–81].

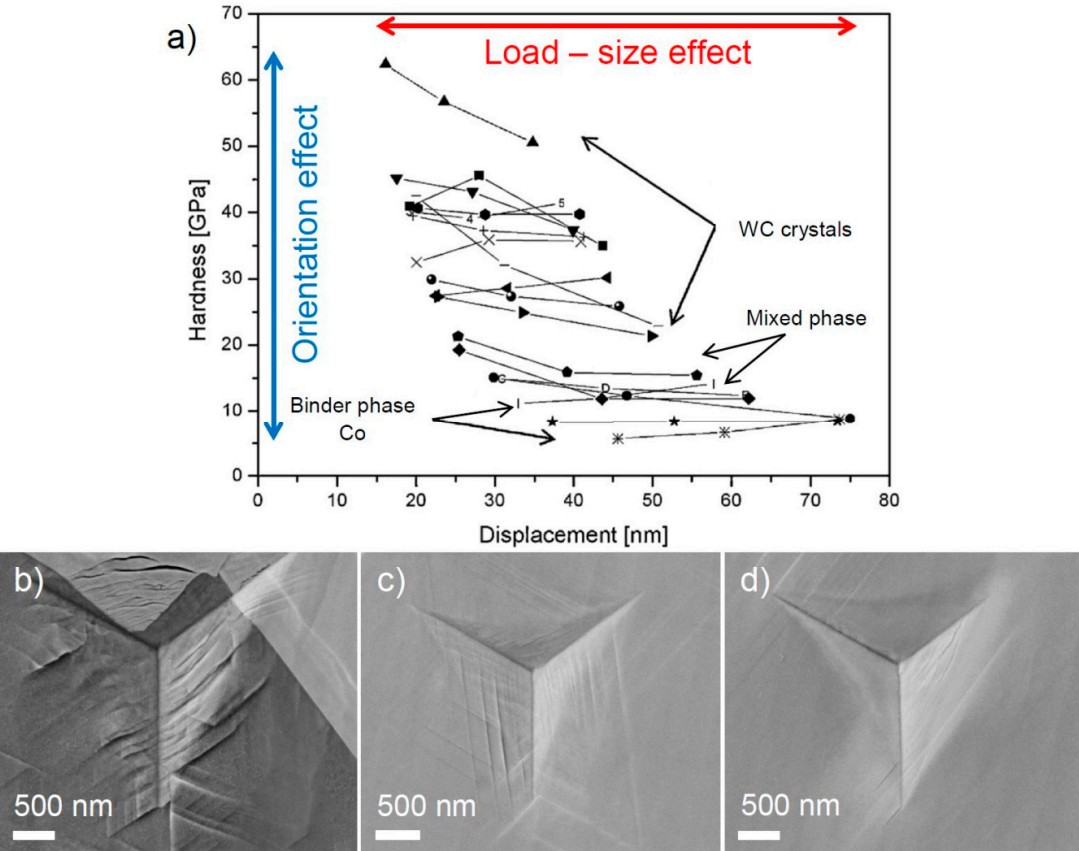

**Figure 5.** (**a**) Load-size effect and orientation effect during continuous multi-cycle indentation of different phases of a WC–Co system at loads of 0.25, 0.5, and 1.0 mN. Connected points represent the same indented constituents of WC–Co under the aforementioned loads. Characteristic impressions in (**b**) mixed-phase of WC grains and Co binder, (**c**) basal, and (**d**) prismatic WC grains, after indentation [71], with copyright permission from Elsevier, 2013.

**Table 1.** Vickers, Knoop, and Berkovich hardnesses reported by various macro-, micro-, and nanoindentation studies on WC single crystals or grains [71–81].

| Author (date), (Ref.) | Form of WC | Type of Indentation | Load Range | Basal Hardness $H$ (GPa) | Prismatic Hardness $H$ (GPa) |
|---|---|---|---|---|---|
| Takahashi and Freise (1965) [72] | single crystal | Vickers macrohardness | 10 N | $H = 22.2 \pm 0.4$ | $H = 11.4 \pm 0.5$ |
| French and Thomas (1965) [73] | single crystal | Knoop microhardness | 1 N | $H = 22–24.6$ | $H = 9.8–23.5$ |
| Pons (1968) [74] | single crystal | Vickers microhardness | 1 N 200 mN | $H = 20.6 \pm 1.1$ $H = 26.5 \pm 1.1$ | $H = 14.4 \pm 1.1$ $H = 15.3 \pm 1.1$ |
| Lee (1983) [75] | single crystal | Knoop macrohardness | 10 N | $H = 19$ | $H = 8–18$ |
| Bonache et al. (2010) [76] | WC grains | Berkovich nanohardness | 0.3–0.9 mN | $H = 25–30$ | $H = 40–55$ |
| Cuadrado et al. (2011) [77] | WC grains | Berkovich nanohardness | 250 mN | $H = 25.6 \pm 0.2$ | $H = 17.2 \pm 0.1$ |
| Roebuck et al. (2012) [78] | WC grains | Vickers microhardness | 200 mN | $H = 23.3$ $H = 32.6$ (AFM) | $H = 14.1$ $H = 21.4$ (AFM) |
| Duszová et al. (2013) [71] | WC grains | Berkovich nanohardness | 10 mN | $H = 40.4 \pm 1.6$ | $H = 32.8 \pm 2.0$ |
| Csanádi et al. (2015) [79] | WC grains | Berkovich nanohardness | 20–25 mN | $H = 43.0 \pm 0.8$ | $H = 28.0 \pm 1.0$ |
| Roa et al. (2015) [80] | WC grains | Berkovich nanohardness | 15–20 mN | $H = 29.9 \pm 4.7$ | $H = 22.0 \pm 9.6$ |
| Roa et al. (2018) [81] | WC grains | Berkovich nanohardness | 4 mN | $H = 32.5 \pm 3.5$ | $H = 25.5 \pm 5.0$ |

With the widespread use of micro/nanoindentation, the mechanical behavior, and anisotropy of micron-scale volumes, such as WC grains, has become a topic of increasing interest for those seeking to understand the basic physical processes from the viewpoint of either materials science or engineering applications [71–81]. Gee et al. [82] were possibly the first to use instrumented nanoindentation for determination of the mechanical properties of the constituent phases of WC–Co on a local scale. They encountered difficulties in the mapping of the individual phases, owing to uncertainties in stage positioning and reliability of software during the measurement, but found it a promising technique for gaining valuable information about the in situ properties of different phases. They reported that, as measured by nanoindentation, the nanohardness of phases of WC–Co cemented carbides and Ti(C,N)–Co cermets is about an order of magnitude higher than the hardness values expected for the constituent phases and the regular hardness of these materials. However, this was actually due to improper calibration of the tip, which was corrected later in the work of Roebuck et al. [78].

Table 1 summarizes the results of the most relevant studies on the effect of crystallographic orientation on the hardness of WC single crystals and WC grains in cemented carbides [71–81]. The majority of measurements were confined to only the two different extremes of orientation and found that the measured hardness values were significantly higher for basal planes than for the prismatic orientation. However, there was one study [76] which found the exact opposite. Their experiment involved the creation of very shallow indents (less than 30 nm depth), where the behavior of the material was possibly dominated by surface effects rather than by the properties of the bulk material. It is important to emphasize that the hardness investigations under discussion were focused on the effect of the basal and prismatic orientations alone.

A Berkovich diamond indenter with loads of up to 0.25 N was used to measure the hardness of individual crystals of WC in a WC–Co system (Cuadrado et al. [77]). Electron backscattering diffraction (EBSD) techniques were used to obtain the orientation of individual crystals and the hardness values were measured for the basal (0001) and prismatic, {10-10} and {1-120} planes of the WC crystals. Hardness values of 20 GPa (basal plane) and 17 GPa (prismatic plane) were obtained.

Roebuck et al. [78] applied depth-sensing microhardness mapping to measure the variation, with applied load and orientation, of the microhardness of WC crystals (of all orientations) of around 50 μm in size embedded in a copper alloy matrix. They found that the most significant effect on the microhardness of WC was found at a deviation angle between the plane of measurement and either the basal or prismatic planes. The grains with a plane close to the basal plane (0001) were found to be considerably harder (approx. 55 GPa at 0.4 N load) than the prismatic ones (approx. 25 GPa at 0.4 N). The hardness values also showed a strong indentation-size effect.

These investigations found the same anisotropy for WC grains as had been reported earlier for WC single crystals (see Table 1).

Systematic experiments have been performed to study the dependence of hardness and deformation mechanism on orientation for WC grains in a large-grained WC–Co hardmetal, using nanoindentation combined with EBSD, SEM, and AFM measurements [79]. The most significant factor affecting the orientation dependence of the nanohardness was the rotational angle ($\Phi$) between the central axis of the investigated hexagonal grain and the surface normal (Figure 6). The nanohardness value was approximately 1.6 times higher on basal ($H = 43$ GPa) than on prismatic planes ($H = 28$ GPa), showing a considerable decrease of around $\Phi = 20$–40 degrees and then practically constant at higher values.

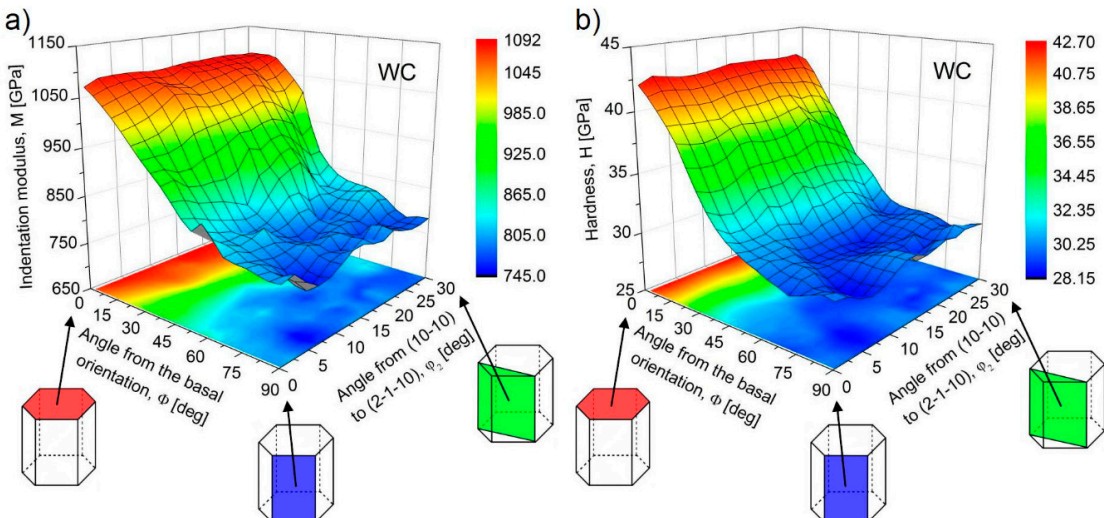

**Figure 6.** The (**a**) indentation modulus and (**b**) hardness of WC grains as a function of the grain orientation in 3-D representation [79].

Massive nanoindentation and statistical analysis of the measured data has provided important information on the small-scale deformation and mechanical properties of WC–Co materials and their constituents [80,81,83]. As in the studies discussed above, a clear and well-defined hardness anisotropy was found for WC crystals, with the hardness values of the prismatic plane and basal plane measured as around 22.0 GPa and 30 GPa, respectively [80]. For the first time, a statistically significant value was reported for the intrinsic hardness of the Co-base binder constrained by the WC grains (without the influence of the surrounding WC particles). This value, 8.1 GPa, was obtained by correcting experimental data for the influence of surrounding carbide grains; this was done by implementing established thin film models. This value accounts for the strengthening of the plastic-constrained metallic phase and is in agreement with values previously estimated in the literature. Such data are critical as input parameters for hardness and toughness modeling, as well as for microstructural design optimization of WC–Co composites.

The small-scale hardness and flow stress of the constrained metallic binder in WC–Co hardmetals with different microstructure parameters have been determined by the help of massive nanoindentation, statistical analysis of the experimental data, and implementation of an established thin film model. The Hall–Petch effect was analyzed on the basis of the mean free path in the binder and a two-phase normalizing parameter was defined as the effective microstructural length scale for studying phase boundary strengthening. It was found that the flow stress of the constrained binder exhibited a linear dependence on the inverse square root of the binder thickness [83].

Further studies improved the accuracy in the determination of the micromechanical properties of each of the constituents of the WC–Co composite. Mean hardness of the cobalt phase (8.0 GPa), WC prismatic plane (25.5 GPa) and WC basal plane (32.5 GPa) were reported [81]. By combining mapping and statistical analysis, attempts were made, for the first time, to determine the mechanical properties at the interface between two differently oriented WC grains and the metallic cobalt binder near the WC/Co interface.

A combined analysis of SEM and the AFM micrographs taken of surface morphology of the basal and prismatic indents, together with the corresponding grain orientations, confirmed that it is mostly the {10-10}<11-23> type slip system that is activated in WC [79]. Additionally, sink-in and pile-up effects were observed in the cases of the basal and prismatic orientations, respectively. EBSD investigation after nanoindentation showed more extensive deformed zones close to the surface around the prismatic imprints than around the basal ones. Considering the operation of the {10-10}<11-23> type slip systems, a theoretical model (easy slip model) was proposed for the measured nanohardness anisotropy, based on the determination of critical resolved shear stress for slip activation. The proposed

theoretical model appropriately describes the measured hardness anisotropy, as shown in Figure 7a. Additionally, the predictions of the model are in good agreement with the characteristics of the observed deformation zones close to the surface of the imprints, especially regarding the sink-in effect. Based on the elastic constants of WC crystals, different models have been used for the prediction of the indentation modulus anisotropy, as shown in Figure 7b. Calculations have revealed that the theoretical (Vlassak–Nix) model is in agreement with numerical (FEM) simulation for a conical tip, while FEM simulation using a Berkovich tip best approximates the experimental results.

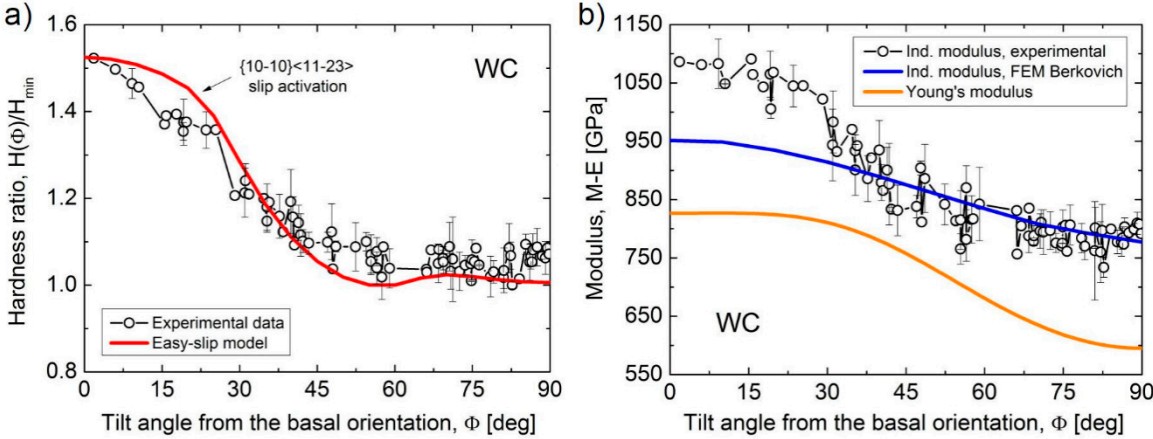

**Figure 7.** Modeling of the orientation dependence of (**a**) hardness, (**b**) indentation, and Young's modulus of WC grains in WC–Co hardmetal [79].

As mentioned, it is not only the hardness of WC grains that is important; their fracture toughness and the hardness of the binder phase (usually Co) also play an important role in the macromechanical behavior, such as hardness and toughness, of hardmetals.

It was found that the fracture toughness of a WC single crystal also varied with the crystallographic orientation. Indentation tests on prismatic planes found a higher toughness value, 8.7 MPam$^{0.5}$, than for the basal one, 7.2 MPam$^{0.5}$. The reason is probably that the prismatic facets have active slip planes that allow for plastic deformation, thus increasing the fracture toughness measured when indenting prismatic planes [77].

In the case of the indentation fatigue test, the indentation depth increased with increasing load more intensively on the prismatic planes than the basal planes. This behavior is probably connected to the different slip and dislocation mechanisms taking place during the indentation of the basal and prismatic planes. This probably results in the generation of sessile dislocations in the basal plane and thus in a higher work-hardening than for the prismatic plane and in different fatigue behavior for the two indented planes [84].

Interesting results have also been reported regarding the hardness of the binder phase. High-resolution nanoindentation rastering of a surface of two grades of WC–Co, using a Berkovich indenter with tip radius of 200 nm and indentation depth of 30 nm, revealed that the Co phase in WC–Co has a significantly higher hardness than measured for bulk Co [85]. The reasons probably involve solid solution of W and C in the Co, differences in thermal expansion between Co and WC, resulting in residual stresses after sintering, and the influence of the hard WC grains on the deformation of the softer binder; differences in the testing procedure must also be considered as a possible factor.

The deformation and mechanical characteristics of submicron-sized binder areas in WC–Co systems are not easy to study, even when using nanoindentation, so a hardmetal binder–like Co-W-C alloy was prepared and investigated [86]. The hardness and elastic modulus exhibited no orientation dependence and also no size effect on the micro/nanometer scales. Twinning was identified as the predominant deformation mechanism during the nanoindentation, with no indication of any

stress-induced phase transformations, probably as a result of the high amount of W and C present in the alloy, which increase the stacking fault energy and stabilize the fcc phase.

EBSD-assisted trace analysis, combined with TEM of indentation-damaged/deformed ultracoarse and ultrafine grain sized WC–Co systems, revealed brittle fractures in the ultrafine system, while the ultracoarse composite exhibited plastic deformation [87]. In the deformed microstructure of the coarse WC grains, there were high densities of dislocations and stacking faults with long sliding distances. More active slip systems were found in the large WC grains with both prismatic and pyramidal characters. In the ultrafine system, the plastic deformation was found mainly in the binder phase, while in the ultracoarse system, dislocations and stacking faults in the WC grains contributed significantly to the plastic deformation and thus to the improved fracture toughness as well.

## 3. Micro/Nano-Tribology and Scratch Testing

In many applications of WC-based hardmetals, wear resistance is a very important property and one which is crucial to optimize at the micro- and nano-levels as well. There is a long history of the examination of the wear behavior of WC-based hardmetals using different techniques, however, micro- and nano-scratch measurements on these systems have been carried out only during the last decade and only by a few authors [88–103].

A cost-effective micro-scratch test system was designed and built at the National Physical Laboratory in Teddington by Gee et al. [88] to meet the following requirements: (i) Flexure suspension such that no bearing friction artefacts are introduced, (ii) ability to carry out experiments in situ using a scanning electron microscope, and (iii) ability to perform single- and multiple-pass experiments remotely. Initial results confirm that the mechanisms of wear that occur in macroscopic contacts also take place in microscale contacts, namely, plastic deformation in both the WC grains and the Co binder phase and cracking in the WC grains.

A series of scratch tests was carried out on WC-based hardmetals with a micro-tribological testing system with diamond indenters of radii from 1 μm to 30 μm and with applied loads from 13.7 mN to 450 mN [89]. To examine scratches made in corrosive media, a droplet of HCl was used. When no acid was present, it was found that the most visible feature of wear was the formation of layers of material on the surface where fragmented WC grains had re-embedded into the binder phase. When acid was present, the binder phase was removed from the surface layers of the hardmetal, leaving the WC grains unsupported. The surface of the hardmetals broke up more rapidly, leaving a highly damaged and fractured surface. Stereoscopic reconstruction was shown to be a valuable way of visualizing and measuring the physical dimensions of scratches, providing a future route for the quantification of damage in these model experiments.

The nanoscratch resistance of WC–Co hardmetals with varying cobalt content and WC grain size has also been systematically investigated [97]. For systems with a high cobalt content, the dominant damage mechanisms are the plastic deformation of the WC grains via slip and the formation of intergranular cracks, which lead to grain fracture. Low cobalt content leads to grains detaching from the surface during scratching at high loads and even during multi-scratch tests at low load. The system with ultrafine WC grain size exhibited very limited slip. The scratch depth and width increased linearly with load, while scratch velocity and load were found to have no significant effect on the scratch friction coefficient. A smaller WC grain size resulted in a lower scratch width and depth and therefore in better scratch resistance.

The influence of the crystal orientation of WC grains in WC–Co-cemented carbide on their nano-scratch resistance has been recently investigated, in a similar manner to the dependence of the hardness [102] (Figure 8). In Figure 8a, characteristic scratch lines, created by 100 mN loads and crossing differently-oriented WC grains, are shown in an AFM micrograph where grains of near-basal and prismatic orientations are marked. The corresponding EBSD map, on which the grain orientations and the scratch directions are visualized, is shown in Figure 8b. Anisotropic deformation characteristics are visible in Figure 8, with different surface morphologies visible in the SEM micrographs taken of the

marked grains with basal and prismatic orientations (Figure 8c,d, respectively). The scratch width is much wider for prismatically-oriented grains than for the grains with basal orientation. Furthermore, parallel slip lines are visible in prismatic orientation, unlike the basal case, where the intersections of the slip planes form triangular reliefs (Figure 8c,d), in agreement with the results of the EBSD measurements. The examination of the observed slip lines suggests that most probably the {10-10}<11-23> type slip system is activated. During the scratch test, numerous cracks were also observed, both inside the WC grains and located at the WC–Co and WC–WC interfaces. On investigation, it was established that the positions of the intragranular cracks were strongly influenced by the grain orientation. Crack lines were visible in the 1010-type prismatic grains. They were perpendicular to the produced slip lines irrespective of the grain position relative to the scratch track, as seen in Figure 8c, where it is clear that the cracks are not perpendicular to the wear track but rather at an angle of ~70°.

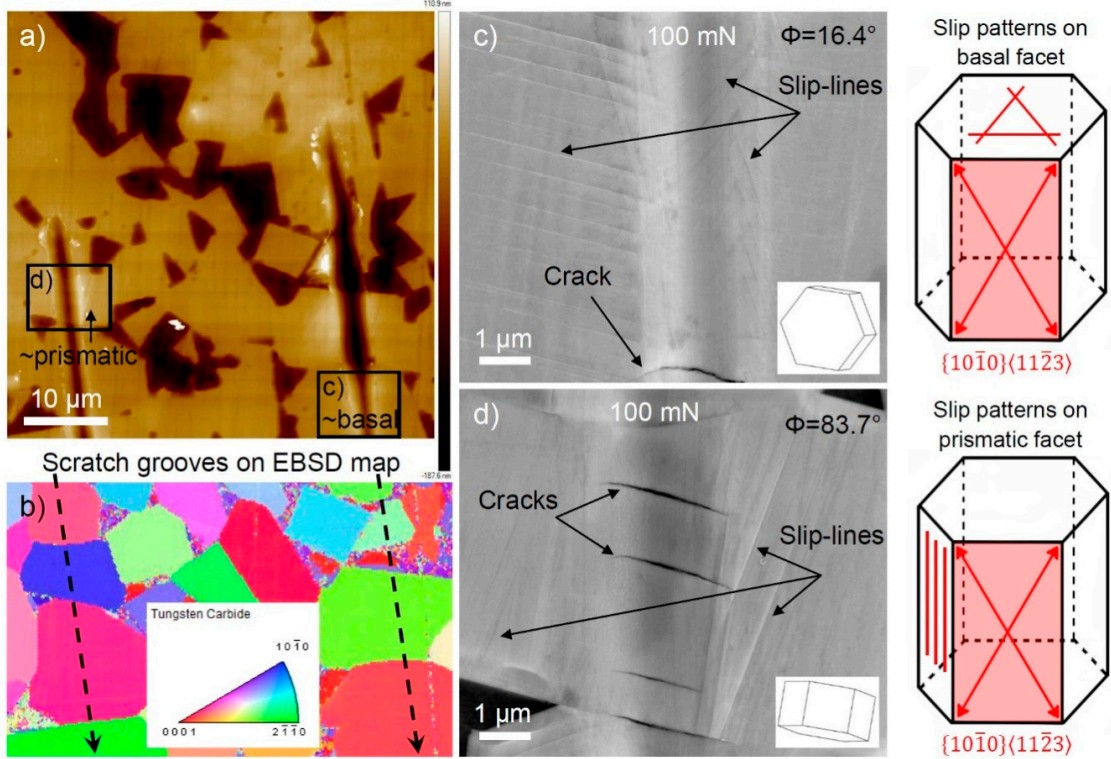

**Figure 8.** Scratch damage characteristics in a WC–Co system: (**a**) AFM micrograph of the characteristic scratch lines crossing differently oriented grains, together with (**b**) the corresponding EBSD image and details of the deformation/damage mechanisms in grains close to (**c**) basal and (**d**) prismatic orientations. Reproduced from [102], with copyright permission from Elsevier, 2015.

The scratch width and depth values were much higher in the grains close to the prismatic orientation than in the grains oriented close to the basal plane, with depth values of approximately 90 nm and 55 nm, respectively. The influence of the grain orientation on the depth of the scratches was similar to that found for the orientation dependence of hardness in the nanoindentation study mentioned earlier, namely, the depth of the scratches increased with increasing angle from the basal orientation (up to an angle of about 40°), showing that the grains became less wear resistant as their hardness decreased. In grains oriented more than 40° from the basal plane, the depths of the scratches were almost constant, just as their hardness remained nearly constant under nanoindentation [79].

The anisotropic scratch resistance of WC grains plays an important role in deformability of WC–Co under tribological conditions. The above-mentioned results are in good agreement with earlier studies regarding scratch tests of WC single crystals, which reported the least brittle behavior during the scratch test of the prismatic plane parallel to the [0001] direction [103]. During the scratch test, the

most brittle behavior was observed on the prismatic plane perpendicular to the [0001] direction and on the basal plane due to the lack of slip planes in the direction of sliding.

In recent experiments, a spherically-tipped diamond indenter was moved under load across polished WC–Co samples of two hardmetal grades with different microstructures, either as a single track or as an abrasion simulation [90]. The effects of changing the number of repeat passes and whether repeat passes were carried out in the same lateral position or with a lateral offset to simulate abrasion were also investigated. The results of EBSD mapping suggest the opposite behavior to that reported in [102]. However, the results were derived from width rather than depth analyses and also on the degree of localized deformation within the local interaction depth (≈50 nm) which generates the EBSD. They found that deformation usually appeared worse in grains which lay only partially in the crack path or where the indenter was moving from a carbide grain into a Co binder (or vice versa).

## 4. Micropillar Compression

There are only a limited number of reports of micropillar testing of WC–Co hardmetals and WC single crystals/grains in their microstructure.

Mechanical, deformation, and failure behaviors of a coarse-grained WC–Co composite, consisting of Co-binder ligaments constrained by the surrounding WC carbide, have been investigated using micropillar compression [104]. Both WC/Co and WC/WC boundaries have been found to be preferential sites for irreversible deformation and failure initiation. Plasticity was found mostly within the softer metallic binder; however, even in this case, deformation takes place in regions adjacent to the carbide–binder interface, where maximum triaxiality stress conditions prevail.

Stress–strain curves showed several strain bursts at different stress levels, ranging from 0.6 to 3.1 GPa. These follow a linear trend as a function of imposed strain and lie between the flow stress values expected for an unconstrained, Co-binder-like model alloy and a highly constrained binder region in bulk WC–Co composites. These results are in excellent agreement with the yield stress values for bulk Co–W–C alloys, which range from 0.4 to 0.8 GPa, depending on W and C additions [105], and with those for the highly constrained binder [12].

A systematic procedure for using FIB milling to produce micropillars of 1 to 4 μm diameter in a WC–Co composite with a small WC grain size was recently reported [106]. As expected, the micropillar with the smallest diameter contained only WC, while micropillars with larger diameters contained WC grains and Co ligaments, with a distribution characteristic of the investigated system. In situ uniaxial micropillar compression and subsequent inspection by field emission scanning electron microscopy (FEMES) were conducted and clear size effects were reported. Smaller specimens with sizes approaching the mean WC size exhibited yielding events (pop-ins) at much higher stresses than the micropillars with sizes of 2 and 4 μm. Such yielding events, between 6.5 and 7 GPa, can be associated with the plastic deformation/failure mechanisms observed for WC grains/crystals with rupture stress above 7 GPa (discussed further below) [102]. On the other hand, for a sample size twice or four times as big as the WC grain size, the mechanical response under uniaxial compression exhibited a combination of several deformation and failure/damage mechanisms, behaving in a manner similar to that of the bulk WC–Co system. The flow stress values of these samples are within the range of the flow stress values reported for highly-constrained Co binder (between 2.2 and 3.7 GPa). Generally, in micropillars, independently of their size, WC-WC or WC–Co interfaces were found to be preferential sites for nucleation of critical damage events and for irreversible deformation and failure phenomena.

Damage analyses of uniaxially compressed WC-(W,Ti,Ta,Nb)C-Co micropillars elucidated the constraining effects on the effective flow stress of the metallic binder, which arose from the physical linking of pop-in events occurring at varying stress levels and deformation/damage micromechanisms taking place in the different constitutive phases [107].

For the understanding of the complex mechanical response of WC–Co systems under uniaxial compression, even at the micro-level, the behavior of the individual constituent phases (WC and binder) is extremely important. There is, at the present time, only one report that has investigated the

mechanical response of differently oriented single-crystal tungsten carbide (WC) micropillars prepared using the FIB process [108]. Depth-sensing indentation and scanning electron microscopy were used for the micromechanical test and damage characterization, respectively. In the case of the micropillars with axes perpendicular to the prismatic plane, the load–displacement curves (Figure 9) exhibited relatively similar slopes, giving a Young's modulus of approximately $E = 660$ GPa. When the yield stress went above $\sigma_y = 6.0$ GPa, the pillars typically deformed very quickly, which resulted in failure of the pillars at stress values above $\sigma_r = 7.0$ GPa. In most cases, the fracture was preceded by a controlled plastic deformation with work hardening and occasionally a sudden strain burst was observed. This stress value is very close to the stress value reported by Sandoval et al. [106] for the micropillar tests when the micropillar was prepared from one WC grain.

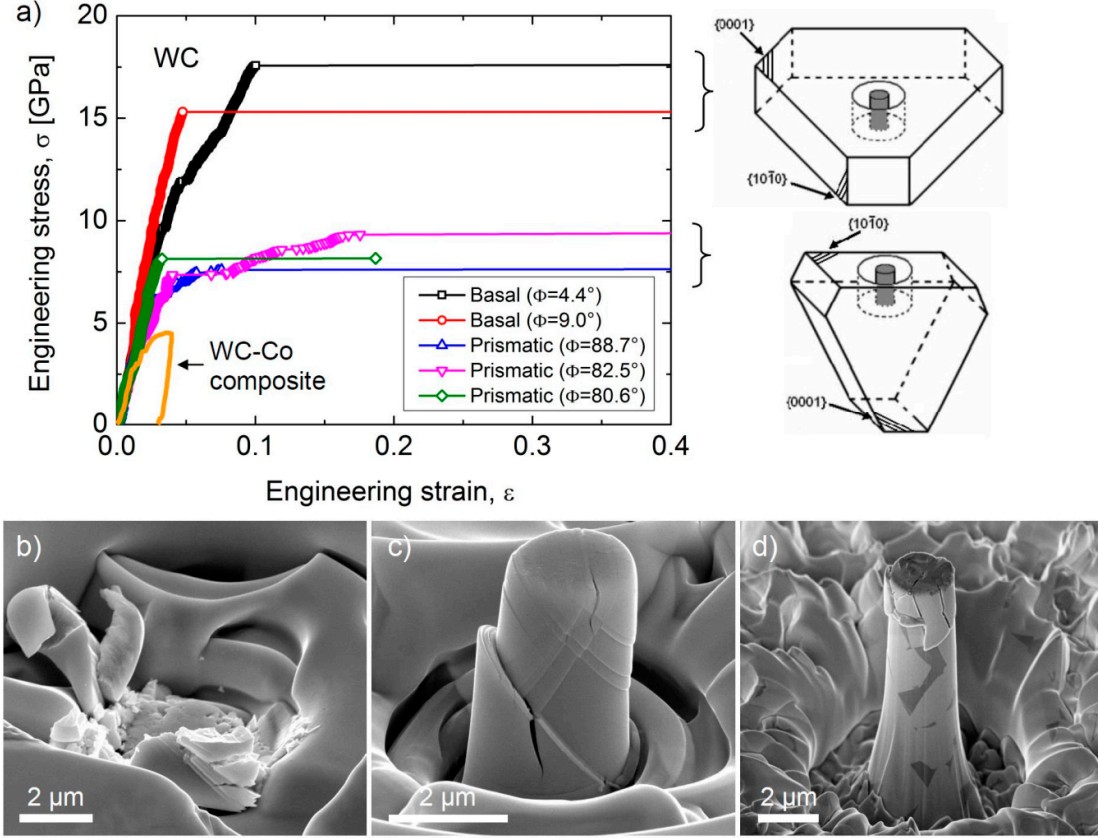

**Figure 9.** (**a**) Stress–strain curves during micro-compression tests of micropillars prepared from WC grains/single crystals with basal or prismatic orientation, together with the stress–strain curve of a micropillar prepared from WC–Co deformed up to ~3.5% and ~4.5 GPa. Deformation and damage mechanisms of the micropillars with (**b**) basal and (**c**) prismatic orientations and (**d**) micropillar containing WC grains and Co binder ligaments. By courtesy of Sandoval et al. [106,108], with copyright permission from Elsevier, 2018.

For micropillars with axes perpendicular to the basal plane, the deformation was brittle with a sudden collapse after reaching the yield/rupture point (elastic limit) at stress values of ~15–20 GPa. The rupture stress was higher when the orientation is closer to the basal direction. Small deviations from linearity were sometimes observed, which suggested low plasticity. This was attributed to such factors as the plasticity of the underlying, softer grains, the presence of the Co phase at the bottom of the micropillar, and the plasticity or fracture initiation inside the micropillar caused by internal defects (e.g., pores).

The main difference between the basal and prismatic orientations is the position of the possible slip systems (Schmid-factor), which resulted in a lower compressive stress/hardness needed for slip

activation for the prismatic orientation than for the basal one, despite the same critical resolved shear stress being necessary in both orientations during micro-compression/indentation.

## 5. Micro-Cantilever Bending and Tensile Test

There are only a few papers which deal with micro-cantilever tests of WC–Co hardmetals and their substitutes and, to the best of our knowledge, only two papers focused on the micro-tensile test of these materials.

The dependence of strength and fracture behavior on specimen size, i.e., the effectively tested volume of two ultrafine grained WC–Co hardmetals, was investigated by varying the effective test volume by over 10 orders of magnitude, including samples with micrometric size [109]. The measured fracture strength values were significantly dependent on specimen size, which can be explained by the well-known volume–size effect. Smaller specimens, with correspondingly smaller effectively tested volumes, also have smaller critical defect sizes. The obtained fracture strength values were in the range of 2.5–6.5 GPa. Specimens with micro-cantilever shapes had strength values of 6.32 and 5.46 GPa (systems with lower and higher cobalt content, respectively). In the specimens with diameters of 3.16 mm, defects such as micro-pores and agglomerates of WC grains larger than the average grain size were identified as origin points of fractures. In the specimen with a diameter of 1 mm, pores and inclusions ranging from a couple of micrometres to sub-micrometre in size were observed.

In the micrometre-sized cantilever specimen, no distinct defects were found as fracture origins. A fracture mechanical estimation of the upper limit for critical defect size in this specimen gave a value close to that of the sub-micrometre-sized tungsten carbide particles. It was therefore expected that this specimen's high tensile strength values of 6.0 GPa were close to the intrinsic strength of the material.

Micro-cantilever in bending was used for testing the mechanical properties of WC–Co hardmetals at microscopic scales for accurate selection of the region at which the fracture occurs, provided that it was located near the beam clamping with the highest bending moment [110]. Both WC grains and single binder phase ligaments were broken during the test and the load-displacement curves were recorded. Fractographic analyses revealed differences in the crack initiation/propagation modes related to specific features of the load vs. displacement curves. From these data, the stresses at which segments of cobalt ligaments and WC grains failed were estimated from linear elastic theory and FEM models. According to these methods, the predicted fracture strength for individual WC grains is between 6.0 and 6.6 GPa.

Recently, micro-cantilever testing was used for the investigation of the fracture strength of different WC/WC interfaces in a WC–Co cemented carbide with large mean WC grain sizes and a very high WC–WC contiguity [111]. In order to distinguish the so-called CSL2-type twist WC/WC interface from other interfaces, the relative orientations were determined by EBSD. The FIB technique was used for machining cantilever beams with a single CSL2 WC/WC interface, located perpendicular to the cantilever axis at a certain distance from the fixed end of the cantilever. Cantilevers with twist boundaries (CSL2) showed a deviation from linear behavior at high deflection values, with small drops/decreases in load just prior to failure. On the other hand, cantilevers with other orientations of WC/WC boundaries showed linear elastic behavior until fracture. Fractographic analyses confirmed that fracture is mainly intragranular in beams with CSL2 boundaries and intergranular for other, non-CSL2 boundaries. Finally, the fracture strengths estimated by LEFM show that CSL2 twist boundaries are much stronger (over 20 GPa) than those of other types (approximately 10 GPa), in agreement with the results obtained by density functional theory—that CSL ($\Sigma$ = 2) twist boundaries are highly coherent and do not allow Co segregation.

A very recent micro-cantilever test aimed at investigating the bending strength and deformability of individual WC grains and WC/WC boundaries [112] revealed that the fracture strength depends very strongly on the fracture origin, as shown in Figure 10. On the basis of this finding, micro-cantilevers were classified into three groups according to the location and character of failure origin:

(1)    Bars from single WC grains without a clear fracture origin, with failure at the fixed end of the cantilever;

(2)    Bars from single WC grains with evident fracture origin in the form of nano-sized defects, with failure further from the fixed end of the cantilever;

(3)    Bars containing a WC/WC boundary as fracture origin with failure at this boundary.

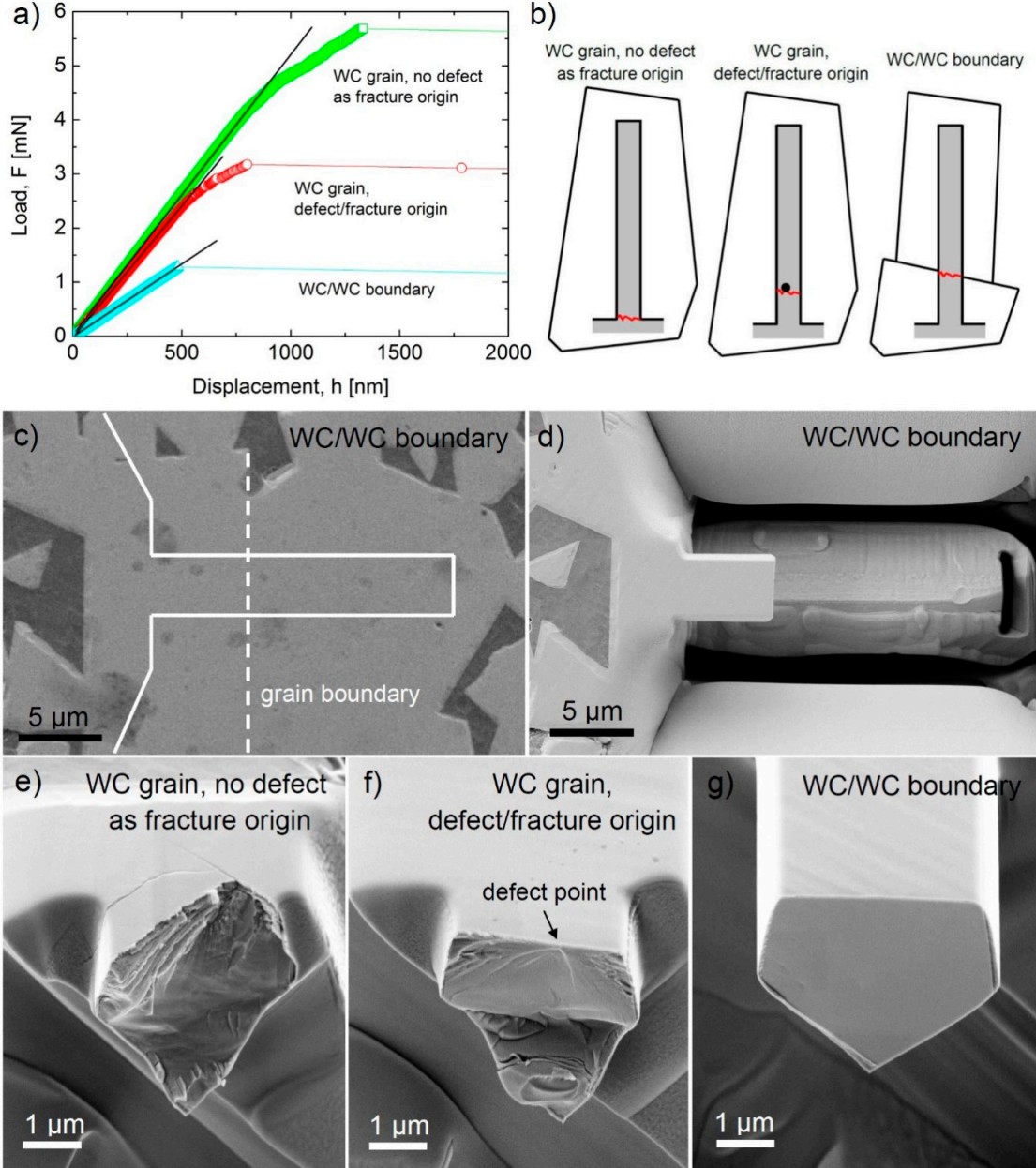

**Figure 10.** Cantilever bending strength testing and characterization: (**a**) Load-displacement curve of typical cantilevers, (**b**) schematic illustration of cantilevers with different fracture origins, (**c**) selection of cantilever position in the microstructure, (**d**) cantilever after bending test. Characteristic fracture surfaces of the cantilevers: (**e**) Fracture in WC without fracture origin/defect, (**f**) with nanometer-sized fracture origin, and (**g**) fracture at the WC/WC boundary [112].

In general, bars fractured on bending in a brittle manner with, in some cases, slight plasticity in the form of slip lines near the main fracture line. Bars without evident fracture origin failed at their fixed end, with a fracture strength from ~10 GPa to ~20 GPa. Load-displacement curves exhibited a deviation from the linear, suggesting a slightly plastic character of the WC grains. Bars containing an identifiable

fracture origin failed at the location of the defect, slightly further from the fixed end, with the presence of the defect acting to increase the stress, and the fracture strength was reduced to approximately 7.5 GPa. In this case as well, slight plastic deformation was also believed to take place before the fracture (see Figure 10a). The fracture origins were clearly visible on the fracture surface, located close to the tensile surface of the bar and approximately 10–15 nm in size (Figure 10f). Micro-cantilevers, containing a WC/WC boundary (or WC/Co/WC/boundary) closer to the fixed end than the half of the beam, broke at this boundary, as indicated in Figure 10c,d. Their fracture strength was found to be the lowest among the three groups of cantilevers, with an average value of 4 GPa. Brittle fracture was observed in most of the tests, leaving a smooth fracture surface after breaking, as shown in Figure 10g. The character of the WC/WC boundaries was not studied during this investigation.

As mentioned, another way to study the micromechanical properties of hardmetals is the micro-tensile testing technique. To the best of our knowledge, there has been only one investigation focusing on the tensile strength and mechanical reliability of WC–Co-cemented carbide nanowires fabricated from the bulk material using the FIB technique [64]. The authors tested 19 nanowires (NW) with various sizes—247–508 nm × 269–1449 nm × 3.7–4.7 μm—using a specially-developed micro-electromechanical systems (MEMS)-based tensile testing device. All nanowires showed linear stress–strain relationships and fractured in a brittle manner without plastic deformation. The Weibull plot of the measured strength values suggested that the fracture mechanisms and origins varied with the size and composition of the nanowires. On the basis of SEM observations, it is possible to say that the dominant failure took place transgranularly through the WC phase if the sample size was small and it consisted only of WC and through the Co binder (or WC/Co interface) in the case of larger samples consisting of WC and Co. The shear fracture strength estimated by measuring the inclined angle of the fractured surface indicated that all the NWs fractured in the shear deformation mode. The apparent Young's moduli ranged from 83.2 GPa to 661 GPa, with the highest value roughly comparable to the nominal value, 550–630 GPa, for WC. The nanowire strength seemed to show a specimen size effect: When the volume was 1.56 μm$^3$, the strength was 1.1 GPa, but on decreasing the volume to 0.53 μm$^3$, the value increased to 5.8 GPa.

An overview of the strength of micro-cantilevers and micro-tensile specimens of WC–Co systems and their constituents with varying effective volume and different fracture origins is given in Figure 11. The results of the four investigations are in relatively good agreement, including the investigation reporting ultra-high bending strength values for WC/WC boundaries without Co content, with values as high as 20–25 GPa [111]. The bending strength of WC grains varied over a range of about 6–21 GPa due to the presence of nano-sized defects as fracture origins, generated during processing, which were more significant than the orientation of WC grains or the error of measurement. The bending of tensile strength of WC/WC boundaries was generally lower than that of single WC grains and ranged from about 2 to 25 GPa. The reason for this wide variation is in strong connection with the type of grain boundaries. For low energy boundaries, like CSL ($\Sigma = 2$) twist boundaries, the connections between the adjacent grains were highly coherent, practically every second atom at the boundary coincided in both lattices. This resulted in a strong bonding between the adjacent grains and high fracture strength that was comparable to bonds in WC grains. For high energy grain boundaries, which was the most common in WC–Co composites, the grain separation was more disordered than that for twist boundaries, with large areas of poor fit. This resulted in a lower bending or tensile strength than that for WC grains. The presence of Co at the WC/WC boundaries weakened boundaries further and also further reduced the fracture strength values, as shown in Figure 11.

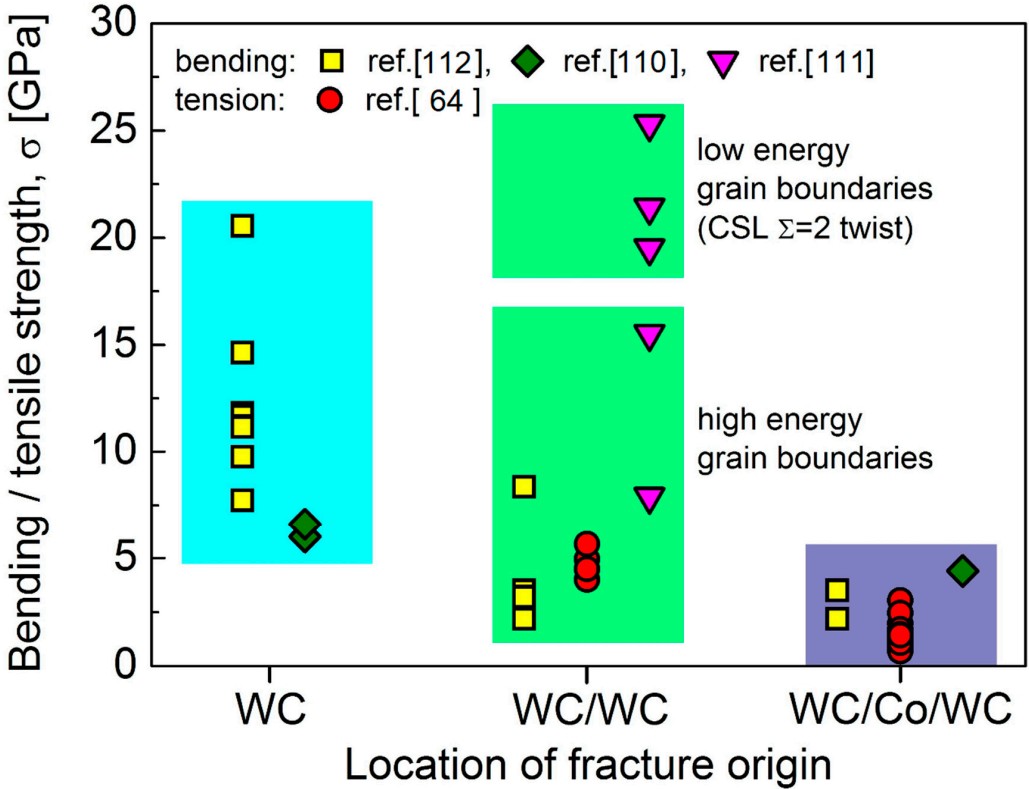

**Figure 11.** Summary of the strength of WC–Co micro-cantilevers and nanowires tested in bending and tension with localized/expected fracture origins [64,110–112].

The deformation and fracture behavior of an ultrafine WC–Co cemented carbide with carbide grain size, approximately 0.6 μm, was investigated using in situ tensile tests in a transmission electron microscope [113]. Two types of samples (thin films with a thickness of approximately 1 μm, prepared by FIB technique), with and without previously introduced microcracks, were tested. In the early stages of loading, the cemented carbide was in an elastic state, but increasing the applied stress caused slip systems to initiate and dislocation motion began to occur in the binder phase. With the increase of dislocation density, work hardening occurred in the binder and dislocations were activated in the WC grains as well. It seemed that for the cemented carbide containing microcracks, coordination of dislocations in WC and Co close to the crack tip was effective at inhibiting crack propagation. These results revealed that the Co binder plays a significant role in the coordination of stress concentration through local plastic deformation, thus protecting the cemented carbide from cleavage fracture.

## 6. Summary and Further Challenges

In this overview, we have summarized the hitherto-published results concerning the small-scale mechanical testing of WC–Co cemented carbides and similar hardmetals, describing the shift from micro- to nanoscale that is one of the most notable trends in the field and looking at the state of the art research in the field of (i) micro- and nanoindentation-induced deformation, fracture/damage characteristics, and related properties (hardness, fatigue, fracture toughness) in cemented carbides and their constituents, (ii) microplasticity and microcrack formation in cemented carbides during scratch testing, (iii) room-temperature deformation of WC–Co hardmetals and WC grains/single crystals during micropillar tests; the specifics of deformation and damage, slip systems, etc., and (iv) strength measurement of individual WC grains and WC/WC grain boundaries using micro-cantilever and tensile specimens.

1.  The micro/nanoindentation tests revealed that the influence of the composition and microstructure parameters of hardmetals on their hardness at the micro level was very similar to that at the macro level. The nanohardness was largely determined by the hardness of the individual phase of the composite under the indenter. Some special micromechanical tests, e.g., the indentation fatigue test, found unusual deformation behavior of WC–Co systems arising from the deformation and damage characteristics of its individual phases. The hardness and indentation modulus of WC grains show clear orientation dependence, with the basal plane showing a significantly higher hardness (approximately 1.4 times higher) than the prismatic one.

2.  Thanks to the progression of high-resolution equipment and methods with good stability and ultra-low drift, small-scale tribological experiments offer new opportunities to investigate the interaction between surfaces and have helped to advance our fundamental understanding of friction, lubrication, and wear of hardmetals at the single asperity contact. Investigation of the influence of the crystal orientation of the WC grains in WC–Co cemented carbide on their nanoscratch resistance discovered a significant anisotropy, with significantly stronger scratch resistance of WC grains oriented close to the basal orientation than for WC grains close to the prismatic orientation.

3.  During micropillar tests, the effect of the scale on the mechanical response of WC–Co composites was clearly connected with the composition and microstructure of the tested micropillars. In the case of micropillars with a number of WC grains and binder areas, the deformation behavior during the compressive test included deformation and damage features at the WC/WC and WC/Co boundaries or in the binder phase. In the case of micropillars prepared from one WC grain (single crystal micropillars), the orientation was found to have a significant influence, with micropillar rupture stress of approximately $\sigma_r = 7.5$ GPa and $\sigma_r = 12.5$ GPa for axes parallel and perpendicular to the basal plane, respectively. The different slip and dislocation mechanisms acting in differently-oriented pillars are probably responsible for this behavior.

4.  The micro-cantilever and micro-tensile strength of WC–Co hardmetals and their constituents is very sensitive to the fracture origin. The strength of WC grains without the fracture origin is around 20–25 GPa; the strength of WC/WC twist boundaries is similar. The strength of the WC grains with nano-sized defect/fracture origin is below 10 GPa and the strengths of the the WC/Co interphase boundary and the Co ligaments are approximately 3 GPa. More investigation is required into the strength of WC/WC boundaries with different WC grain orientations and the strength of the WC/Co/WC boundaries.

In conclusion, this overview of the past decade's research into small-scale mechanical testing of hardmetals shows that the current testing/characterization methods provide suitable tools for obtaining new and fundamental insight into the mechanical properties, deformation, damage, and fracture mechanisms of these composites and their constituents at the micro/nano-level.

*Further Challenges*

A.  Optimization of experimental conditions: (i) Mechanically-polished surface preparation connected with nanoindentation and scratch tests; (ii) damage-free FIB-milled specimen preparation (micropillar, cantilevers, and tensile samples); (iii) misalignment during the micropillar and micro-cantilever test; (iv) different testing rates and modes (fatigue, impact, etc.); (v) tests at high temperatures—indenter tip, etc.; (vi) in situ testing in combination with analytical units.

B.  Effect of the microstructure parameters on (i) deformation, damage, and fracture phenomenon during micro-indentation and micropillar compression and (ii) deformation and damage evolution during micro/nano-scratching and tribology.

C.    Size and orientation effect of constituent phases: (i) Indentation size effect during the micro- and nanohardness testing of different carbide and binder phases; (ii) effect of size/diameter of micropillars and crystal orientation during micro-compression on slip activation, deformation mechanisms, yield and rupture strength of carbide phases; (iii) effect of cantilever size and crystal orientation on bending strength, Young's modulus, fracture toughness, etc.; (iv) effect of the crystal orientation of neighboring carbide grains on carbide/carbide interphase fracture and fatigue strength during micro-cantilever test.

D.    Loading rate/mode and temperature effect on deformation and damage characteristics of (i) micro-sized bulk hardmetals, (ii) their constituents, and (iii) their interphases.

E.    Modeling—by way of density functional theory (DFT) calculation, discrete dislocation dynamics modeling, etc.—of the observed phenomena concerning the deformation, damage, and fracture mechanisms in different hardmetals, in order to assist the design and development of new systems with an optimal combination of mechanical and tribological properties.

**Acknowledgments:** The authors gratefully acknowledge the financial support from the POWROTY/2016-1/3 project as carried out within the Powroty/Reintegration programme of the Foundation for Polish Science co-financed by the European Union under the European Regional Development Fund. The part of work was financed by the projects: APVV-15-0014 (ProCor), VEGA 2/0130/17, VEGA 1/0096/18, and "Research Centre of Advanced Materials and Technologies for Recent and Future Applications PROMATECH", ITMS 26220220186, supported by the Operational Program "Research and Development" financed through European Regional Development Fund.

**Conflicts of Interest:** The authors declare no conflict of interest.

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
