# Peer review of "Small-Scale Mechanical Testing of Cemented Carbides from the Micro- to the Nano-Level: A Review"

_metals, doi:10.3390/met9050502_

Reviewer 1 Report

- This is not a research paper but a review. It should include a critical analysis of the state of the art. Further analysis and explanation of the goals accomplished is needed along with a clear identification of future challenges of these technologies.

- The authors include some of their unpublished work as a reference. If it is not published yet, it cannot be included in the review (by definition).

Author Response

The authors much appreciate the detailed and helpful comments of the Reviewers. The paper has been revised to address the points raised by the Reviewers and the amended sections are marked using yellow highlight. Specific responses to the Reviewers’ comments are given below (the Reviewers’ comments are in normal style while the answers are in bold).

Responses:

We respect the Reviewer’s comments but do not agree completely the argument. The present review paper summarizes the most relevant works, or some in cases all of them that have been published up to now, in the field of micro/nanomechanical testing of WC-Co hardmetals, including micro/nanoindentation, micro/nanotribology and scratch testing, micropillar compression and micro-cantilever bending and tensile tests. In each testing method, results are not only described but also compared with each other and the common main deformation mechanisms are highlighted. In order to shed light on the deformation process and/or resolve the paradox between some works, the authors inserted indeed their unpublished results only in one part in the form of Figure 11, which is thought to be helpful for the reader. Although Figure 11 itself is unpublished yet, all the data points are published in ref. [111].

Reviewer 2 Report

Interesting overview of the topic. 

A few minor remarks:

Lines 75-77: It seems that a verb is missing in this sentence.

Line 180: Remove "of".

Line 214: H2 with 2 as subscript.

Line 218: "are not behave" change to "don't behave".

Line 504: "plain" change to "plane".

Line 521: Punctuation sign missing after [108].

Line 639: "lloking" change to "looking".

Line 650: Remove "Do".

Line 661: "a discovered" change to "discovered a".

Line 685: Remove underscore under "i)".

Line 702: "Density Functional Theory" instead of "Density Function Theory".

Author Response

The authors much appreciate the detailed and helpful comments of the Reviewers. The paper has been revised to address the points raised by the Reviewers and the amended sections are marked using yellow highlight. Specific responses to the Reviewers’ comments are given below (the Reviewers’ comments are in normal style while the answers are in bold).

Response:

The authors thank for the Reviewer’s comments and suggestions. Corrections have been made in the manuscript as indicated by yellow highlights.

Reviewer 3 Report

The reviewer finds this review article interesting. It is well structured and covers mostly very recent publications.

A couple of comments:

The introduction section is too long.

Caption texts of some figures such as Fig. 3, 5, 9 and 10 are too long. Consider to move the explanations.

The texts in Fig. 11 are too large compared with the font size of the body text.

Line 639: "... and lloking at the state ... " --> "... and looking at the state ...." 

Author Response

The authors much appreciate the detailed and helpful comments of the Reviewers. The paper has been revised to address the points raised by the Reviewers and the amended sections are marked using yellow highlight. Specific responses to the Reviewers’ comments are given below (the Reviewers’ comments are in normal style while the answers are in bold).

Response:

The authors thank for the Reviewer’s comments and suggestions. Corrections have been made in the manuscript as indicated by yellow highlights.

1. Length of the Introduction, in our opinion it is adequate for the type of paper we had in mind.

2. We think the captions should convey the ideas illustrated clearly.

3. We find the letters to be of a size which makes them clearly readable.

Reviewer 4 Report

In its present form, the manuscript contains the following sections:

Abstract

1. Introduction

2. Micro/Nano-Indentation

3. Micro/Nano-Tribology and Scratch Testing

4. Micropillar Compression

5. Micro-Cantilever Bending and Tensile Test

6. Summary and Further Challenges

Acknowledgments

References (with a total of 113 references).

The structure of this review article is acceptable, and the review is of interest to the readers, but before publication some improvements or corrections must be made:

Line 24: Correct “background”.

Lines 39, 40, 41: The following list of statements “The bending strength increases with decreasing flaw size, the fracture toughness with increasing mean free path in the binder phase, and the hardness with increasing contiguity (increasing WC/WC interface)” appears here (at the Introduction) without clear indication of the References that support those statements. Readers would like to understand what is the meaning of “mean free path in the binder phase” and what is the right meaning of “contiguity”.

Line 73: Explain to the readers the meaning of HEC. Is it High-Entropy Carbide (HEC) ?

Lines 75, 76, 77: Please note that there is not any verb in the whole paragraph.

Line 94: Note that Figures 1c) and 1d) just show F and not Fmax.

Line 95: In order to use the same format as in Figure 1c) it is preferable to write Fmax > 2 N .

Line 147: To help the readers, please provide the explanation/meaning of the acronym LIGA.

Figure 3: Note that the values on both axes of Figure 3a) need to be legible.

Line 180: The phrase needs correction.

Line 183: To help the readers, it would be important to explain what is the meaning of “indentation modulus. (Note that “indention modulus” appears here in the text for the first time).

Line 214: What is the meaning of H2? It should be H2.

Line 218: English phrasing needs revision.

Line 231: What is the meaning of EIt ?

Line 263: Cuadrado et al. is reference [76] instead of [10].

Line 265: (1-120) planes ??? And why do you not use the family symbol for planes, e.g. {10-10} ?

Table 1: Instead of just writing [71-80] in the table title, it is more adequate to indicate the reference close to each authors’ name or adding another column to the table.

Line 342: It seems that the authors have adopted the UK English instead of the US English (e.g. previously they use “behaviour” instead of “behavior”). Then, please write the whole text in good English (UK or US usage is accepted, but not a mixture of these). Other occurrences of “behavior” are highlighted in the marked version (send as an attachment).

Line 443: Is it necessary to use the symbol S ?

Line 447: Is it correct to use 50 nm has an indication of volume?

Line 624: strength

Line 650: Do Some ???

Line 684: Before presenting the list of “further challenges”, perhaps it would be adequate to write a sentence explaining what the authors mean by “further challenges”. Also, it seems adequate to attribute a letter to each of the “further challenges”:  A. Optimization of experimental conditions: i) mechanically-polished … …  B. Effect of the microstructure parameters on … … C. Size and orientation effect of constituent phases: … … D. Loading rate/mode and temperature effect on … … etc.

Author Response

The authors are grateful for the thorough review. The entire manuscript has been corrected according to the Reviewer’s comments and suggestions, indicated by yellow highlights.

Round  2

Reviewer 1 Report

It is claimed that results from different authors are discussed, but these results are just repeated and not discussed.

There are no clear conclusions coming from the authors which add some insight to the works already published.

There are many Figures that have been removed in this version. Why is this so?

Author Response

The authors much appreciate the detailed and helpful comments of the Reviewers. The paper has been revised to address the points raised by the Reviewers and the amended sections are marked using yellow highlight. Specific responses to the Reviewers’ comments are given below (the Reviewers’ comments are in normal style while the answers are in bold).

Comments and Suggestions for Authors:

1) It is claimed that results from different authors are discussed, but these results are just repeated and not discussed.

Response: We think so that along with the description of the different results their common physical explanation is also mentioned, considering as a discussion. For example in nanoindentation, it was done in lines 320-335 using the Authors’ earlier work [78].

2) There are no clear conclusions coming from the authors which add some insight to the works already published.

Response: Respectfully but we don’t agree with that. In the conclusions, it is highlighted that how important is the anisotropy of WC grains during micropillar compression, nanoindentation, etc., and how sensitive is the bending strength to the defect structure over other effect (e.g. anisotropy) during micro-cantilever bending.

3) There are many Figures that have been removed in this version. Why is this so?

Response: None of the figures have been removed from the revised manuscript. Please, download the original and revised versions and compare them.

Reviewer 4 Report

This revised version (metals-478843-peer-review-v2) is now practically ready for publication. I just noticed two situations that can be improved:

1) Line 101 (caption of Figure 1): In order to use the same format as in Figure 1c) it is preferable to write: Fmax > 2 N (instead of: 2 N ≤ Fmax ).

2) Line 234: For increasing English style and elegance, the phrase “According to the results these do not behave according to this theory” needs revision.

Author Response

Comments and Suggestions for Authors:

This revised version (metals-478843-peer-review-v2) is now practically ready for publication. I just noticed two situations that can be improved:

1) Line 101 (caption of Figure 1): In order to use the same format as in Figure 1c) it is preferable to write: Fmax > 2 N (instead of: 2 N ≤ Fmax ).

2) Line 234: For increasing English style and elegance, the phrase “According to the results these do not behave according to this theory” needs revision.

Responses: Thank you for the comments. Corrections have been made in the caption of Fig. 1 and in line 221, highlighted by yellow.